# Influence of N6-methyladenosine (m6A) modification on cell phenotype in Alzheimer's disease

**Pengyun Ni** [1]*, **Kaiting Pan** [2], **Bingbing Zhao** [3]

**1** Department of Science and Education, Baoji Traditional Chinese Medicine Hospital, Baoji, Shannxi, P.R China, **2** Department of Neurology, Baoji Third Hospital, Baoji, Shannxi, P.R China, **3** Emergency Department, Baoji Traditional Chinese Medicine Hospital, Baoji, Shannxi, P.R China

These authors contributed equally to this work.

* 469079448@qq.com

**Data Availability Statement:** All Expressions of GSE5281 and GSE48350, gene of m6A datas are available from the Figshare database. (DOI: 10.6084/m9.figshare.22656136, URL: https://figshare.com/account/home).

## Abstract

### Objective

Recent research has suggested that m6A modification takes on critical significance to Neurodegeneration. As indicated by the genome-wide map of m6A mRNA, genes in Alzheimer's disease model achieved significant m6A methylation. This study aimed to investigate the hub gene and pathway of m6A modification in the pathogenesis of AD. Moreover, possible brain regions with higher gene expression levels and compounds exerting potential therapeutic effects were identified. Thus, this study can provide a novel idea to explore the treatment of AD.

### Methods

Differential expression genes (DEGs) of GSE5281 and GSE48350 from the Gene Expression Omnibus (GEO) database were screened using the Limma package. Next, the enrichment analysis was conducted on the screened DEGs. Moreover, the functional annotation was given for N6-methyladenosine (m6A) modification gene. The protein-protein interaction network (PPI) analysis and the visualization analysis were conducted using STRING and Cytoscape. The hub gene was identified using CytoHubba. The expression levels of Hub genes in different regions of brain tissue were analyzed based on Human Protein Atlas (HPA) database and Bgee database. Subsequently, the candidate drugs targeting hub genes were screened using cMAP.

### Results

A total of 42 m6A modified genes were identified in AD (20 up-regulated and 22 down-regulated genes). The above-described genes played a certain role in biological processes (e.g., retinoic acid, DNA damage response and cysteine-type endopeptidase activity), cellular components (e.g., mitochondrial protein complex), and molecular functions (e.g., RNA methyltransferase activity and ubiquitin protein ligase). KEGG results suggested that the above-mentioned genes were primarily involved in the Hippo signaling pathway of

**Funding:** The authors received no specific funding for this work.

**Competing interests:** The authors have declared that no competing interests exist.

neurodegeneration disease. A total of 10 hub genes were screened using the protein-protein interaction network, and the expression of hub genes in different regions of human brain was studied. Furthermore, 10 compounds with potential therapeutic effects on AD were predicted.

## Conclusion

This study revealed the potential role of the m6A modification gene in Alzheimer's disease through the bioinformatics analysis. The biological changes may be correlated with retinoic acid, DNA damage response and cysteine-type endopeptidase activity, which may occur through Hippo signaling pathway. The hub genes (SOX2, KLF4, ITGB4, CD44, MSX1, YAP1, AQP1, EGR2, YWHAZ and TFAP2C) and potential drugs may provide novel research directions for future prognosis and precise treatment.

## Introduction

Alzheimer's disease (AD) has been confirmed as one of the most common Neurodegeneration, characterized by a progressive decline in memory function and even cognitive impairment [1]. AD has a high incidence in the elderly population and in some countries [2]. Besides, AD is recognized as a problem that should arouse the attention of the society and the government. Due to the lack of effective treatment, AD patients and their families are currently subjected to considerable number of inconvenience and health risks. The pathogenesis and therapy of AD should be urgently studied.

N6-methyladenosine (m6A) refers to a common reversible methylation modification, including methylation, demethylation, and recognition of RNA Post-transcriptional modification [3]. Existing research has confirmed that m6A extensively exists in mammals, with an average of 3–5 M6A sites per mRNA transcript on isolated RNA adenosine [4]. m6A is capable of disrupting chromatin-modifying enzyme-associated transcript stability and thereby regulate histone modifications [5]. m6A influences mRNA metabolism by regulating RNA degradation [6], translation [7], RNA splicing [8] and nuclear export [9] through multiple types of protein complexes. In animals, known methyltransferase protein complexes include methyltransferase protein 3 (METTL3) and METTL14, which interact with METTL14 to form heterodimers that exert methyltransferase activity [10–12]. m6A removal is mainly accomplished by removal of the protein AlkB homologue 5(Alkbh5) and the obesity-associated protein FTO [10, 13]. The decoding of m6A labeling is performed by YTHDF1/2/3, YTHDC1/2, hnRNP a 2/B 1 and RNA binding proteins [6, 10, 13–17].

m6A refers to a reversible RNA modification [18] occurring in the brain [19], and it plays a certain role in the physiological functions of neurogenesis [5], learning and memory [20]. Existing research has suggested that mutations in a wide variety of m6A participants are correlated with neurological disorders [21]. The methylation level of m6A was significantly increased in the AD model, suggesting that m6A may play a key role in AD [22]. However, the mechanism of m6A in the brain is still not perfect and needs further study.

In this study, the data were collected from public databases, and bioinformatics methods were adopted to explore m6A-modified hub genes and pathways in Alzheimer's disease. This study aimed to further clarify the expression of key genes in a wide variety of brain regions and predict potential therapeutic compounds, so as to explore novel ideas and lay a research basis for the treatment of AD.

## Materials and methods

### Data acquisition and processing

Two microarray datasets (GSE5281 and GSE48350) were downloaded in this study from the Gene Expression Omnibus (GEO) database [23] (http://www.ncbi.nlm.nih.gov/gds/) investigating gene expression profiles changes in patients with Alzheimer's disease.

The gene expression profile GSE5281 and GSE48350 were generated on the platform of GPL570 [HG-U133_Plus_2] Affymetrix Human Genome U133 Plus 2.0 Array. This dataset comprises 247 control samples and 167 AD samples.

### Analysis of differentially expressed genes (DEGs)

GSE5281 and GSE48350 were downloaded from the GEO database using the GEO query package [24] to remove one probe for multiple molecules. When a probe corresponding to the same molecule is encountered, only the probe with the largest signal value will be retained. Subsequently, Limma package [25] was adopted to analyze the difference between the two groups. Volcano plots were drawn using fold change and corrected p-values. |log2(Fold Change)| > 0.58 and P.value < 0.05 were considered the threshold for the DEGs. The expression heatmaps were exhibited using the R package pheatmap [26].

### N6-methyladenosine (m6A) modification gene and Venn analysis

The GeneCards database [27] (https://www.Genecards.org/) was searched for relevant genes with the keywords "m6A" and "N6-methyladenosine", and the organisms were set as "homo sapiens". Subsequently, the Venn diagram between AD up-regulated and down-regulated DEGs and m6A genes was generated using Bioinformatics (http://www.bioinformatics.com.cn/). Next, the overlapping genes of DEGs and m6A were analyzed.

Next, the expression differences of the overlapping genes between the AD group and the control group were visualized using PCA maps. On that basis, the individuals of up-regulated gene and down-regulated gene expression was proved.

### Gene ontology (GO) enrichment analysis and and Kyoto encyclopedia of genes and genomes (KEGG) pathway analysis

Metascape [28] (https://Metascape.org/gp/index.html#/main/step1) and Bioinformatics were adopted to study the enrichment of overlapping genes by process and pathway. The GO terms for Biological Process (BP), Cellular Component (CC), and Molecular Function (MF) categories were enriched using the Metascape online tool. The biological processes, cellular components, molecular functions and KEGG pathway enrichment analysis of the overlapping genes were identified with the human genome as a background reference and P value <0.05 as the inclusion criteria.

### PPI network construction and module analysis

The STRING database (https://cn.string-db.org/) refers to a tool for assessing functional protein-protein interactions (PPI) [29]. A PPI network of overlapping genes was built on STRING 11.5 to predict interactions between proteins encoded by genes that take on critical significance to m6A modification in AD. Subsequently, the PPI data was imported into Cytoscape 3.7.2 software [30] for visualization, and the Hub gene was screened out using the CytoHubba [31] plug-in.

### Selection and analysis of hub genes

The top 10 core genes were screened using CytoHubba, a plugin in Cytoscape software. The Protein expression profiles of hub genes on the tissue level and the gene expression level originated from the HPA (Human Protein Atlas) database. The gene expression scores of hub genes on the tissue level originated from the Bgee database (https://bgee.org).

### Identification and verification of hub genes

We brought hub genes into the GSE5281 and GSE48350 datasets to identify and validate hub genes expression levels and diagnostic value. We visualized the gene expression levels of the hub genes between AD patients and healthy controls using a Box graph and Scatter plot. AUCs were calculated by receiver operating characteristic (ROC) curve analysis to determine the predictive value of hub genes.

### Screening for potential pharmacological targets

In this study, compounds with potential therapeutic effects on AD were predicted using the cMAP database [32] (ConnectivityMap, https://clue.io/query). The CMAP database can be adopted for the comparison of similarity between drug-induced gene profiles and gene expression. Scores over 0 represent the small-molecule compounds similar to those for monitoring gene changes, while scores less than 0 represent the small-molecule compounds with opposite effects on genes to be tested that may exert therapeutic effects on disease. Screening for compounds with connectivity score less than -80 is recognized as a reliable prediction. Through the comparison of the Gene expression profiling data between the compounds and cell lines, compounds with fractions less than 0 were identified as having potential therapeutic effects on AD. Thus, as indicated by the comparison of the Gene expression profiling data between the compounds and cell lines, compounds with a score of less than -80 that may have potential therapeutic effects on AD should be screened.

## Result

### Identification of DEGs during AD

The differential gene expression analysis between samples was performed to compare the difference in gene expression between the AD group and the control group. The gene dataset GSE5281 and GSE48350 contained a total of 167 Alzheimer's disease samples and 247 normal samples. With the Limma package for differential expression analysis, with $|log2(Fold Change)| > 0.58$ and P.value $< 0.05$ as the screening conditions, 917 differently expressed genes in brain tissues of AD patients were identified, as compared with normal brain tissues. To be specific, 265 were up-regulated, and 652 were down-regulated, and the clustering analysis of the above-described differential genes was conducted, with the result illustrated in the volcano plot (Fig 1A). Fig 1B presents a heat map of data set, suggesting that the clustering of samples is highly reliable.

### M6A modification gene and Venn analysis

In this study, 1126 m6A genes were identified by searching GeneCards and BioGPS databases. AD up-regulated and down-regulated DEGs gene and m6A gene were introduced into the Venn analysis tool. There were 42 overlapping genes, comprising 20 up-regulated overlapping genes and down-regulated 22 overlapping genes (Fig 2A). Furthermore, as depicted in the PCA plot, the control group and AD group samples were separated, suggesting that the

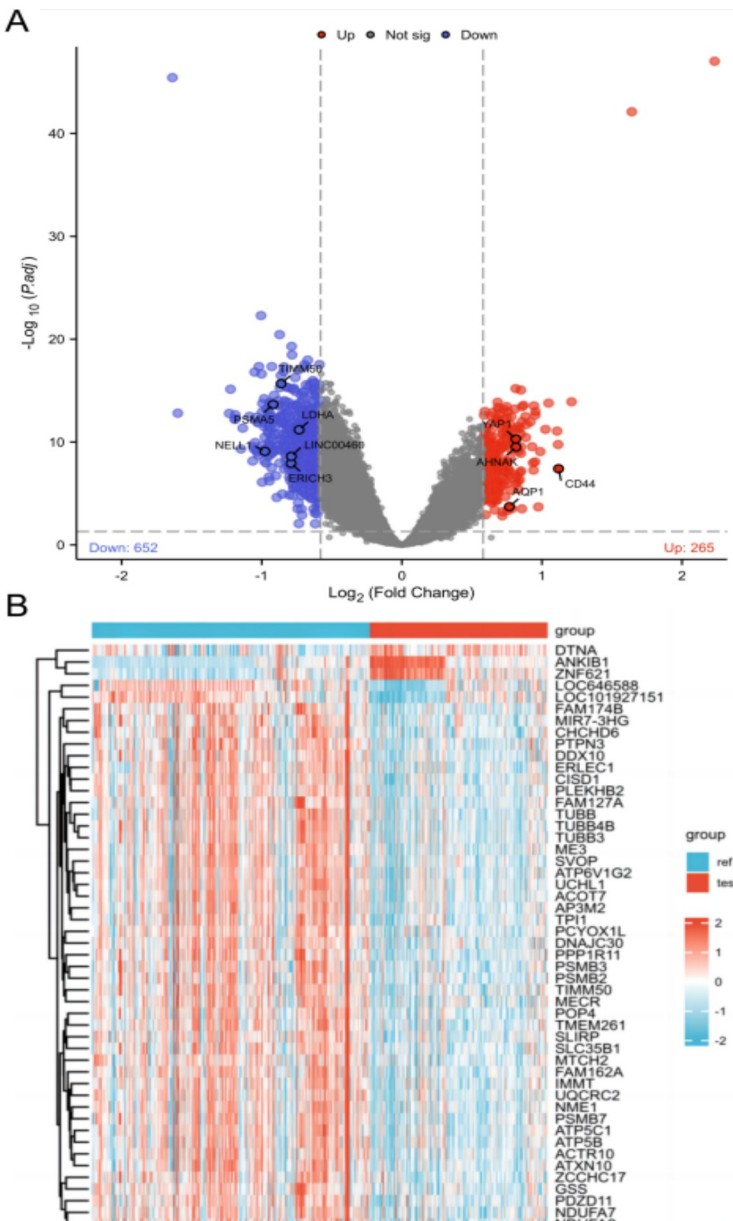

**Fig 1. Differential expressed gene (DEG) analysis on Alzheimer's disease.** (A) Volcanic plots of gene expression of Alzheimer's disease in GSE5281 and GSE48350. Red represents upregulated DEGs, blue represents downregulated DEGs. P < 0.5, |log2(Fold Change)| > 0.58. (B) Heat map of the DEGs. Showing the 50 differentially expressed genes. Red color represents up-regulated genes, and blue represents down-regulated genes.

expression of up-regulated and down-regulated overlapping genes different between the AD Group and the control group (Fig 2B).

## GO and KEGG Pathway enrichment analysis

The GO and KEGG enrichment analysis was conducted on the overlapping genes to study the function of the overlapping genes of AD DEGs and m6A gene. As indicated by the GO

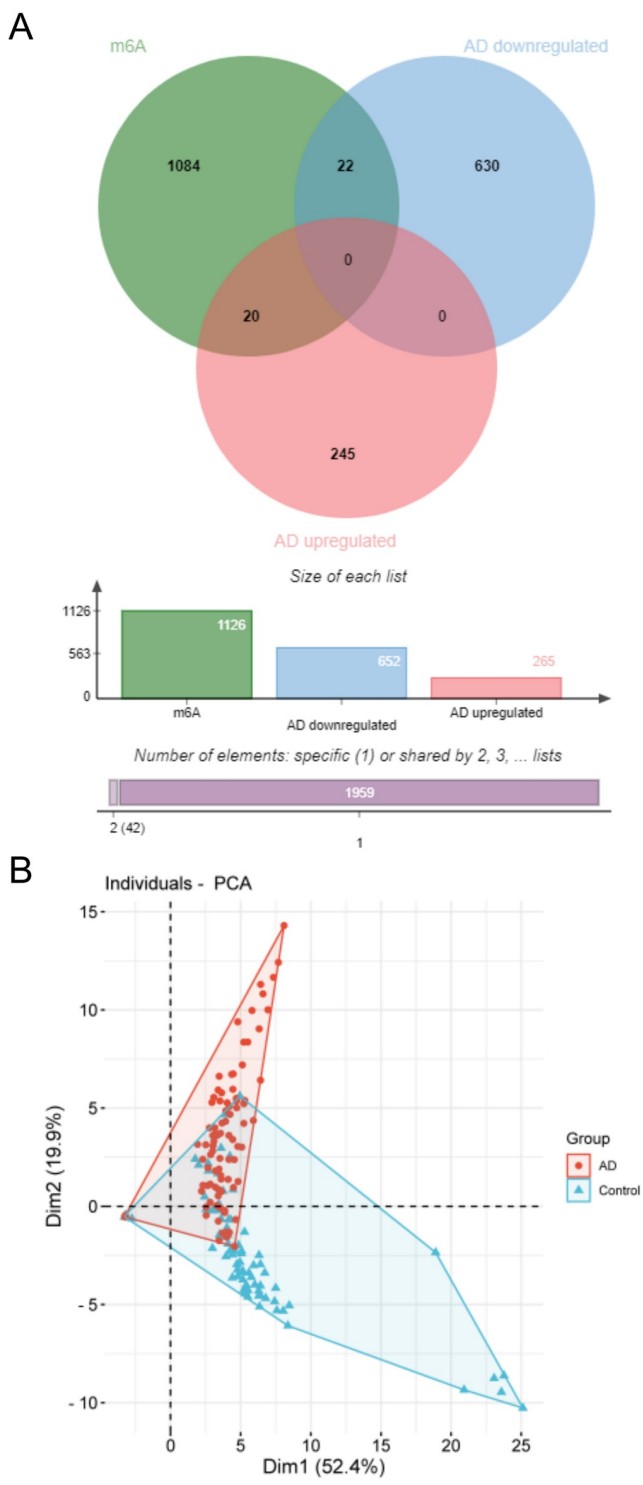

**Fig 2. Overlapping genes.** (A) Venn diagram of the overlapping genes of AD DEGs and m6A gene. Red represents upregulated DEGs, blue represents downregulated DEGs. (B) PCA indicates that the control group and AD group samples were separated, suggesting that there were differences in gene expression of up-regulated and down-regulated overlapping genes different between the groups.

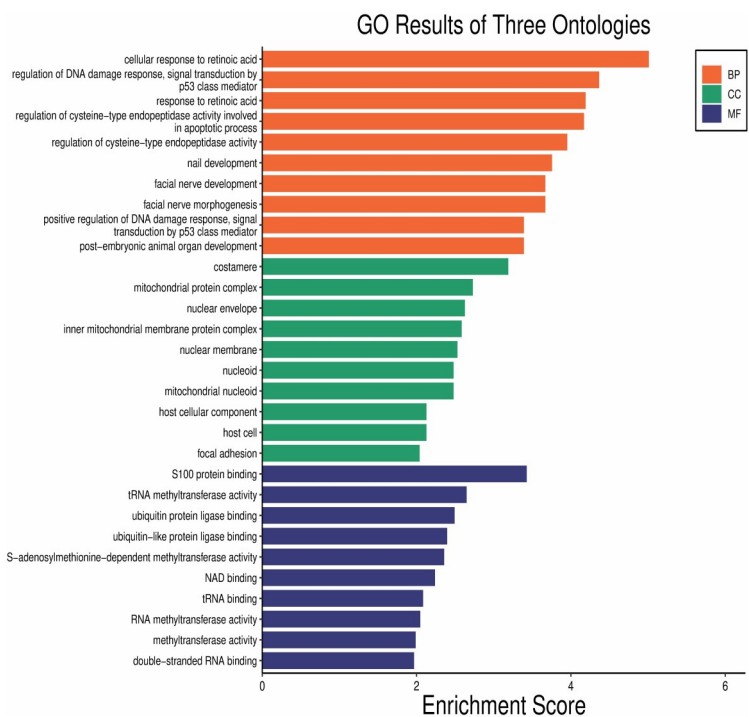

**Fig 3. GO analysis of the overlapping genes of AD DGEs and m6A genes.** The bars show the Top10 processes enriched by overlapping genes in BP, CC and MF.

analysis results, the overlapping genes were primarily enriched in involving in retinoic acid, DNA damage response, and cysteine-type endopeptidase activity of BP, mitochondrial protein complex of CC, and RNA methyltransferase activity and ubiquitin protein ligase of MF (Fig 3). The top 10 processes with the smallest P-value of three ontologies of BPs, CCs, and MFs enriched by AQP1, MSX1, ITGB4, ADARB1, EEF1E1, KLF4, FXR1, TRMT10C, UQCRC2, NUP93, LRPPRC, AHNAK, ATP2A2, YWHAZ, IDH3G, NDUFA13, CCT2, MECOM, PDK4, SOX2, YAP1, CD44, EGR2, LDHA, TIMM50, NELL1, NSUN6, as well as PSMA5 were enriched in three aspects of BPs, CCs and MFs (Fig 4A–4C). The KEGG result indicated the overlapping genes enriched in neurodegeneration disease and Hippo signaling pathway (Fig 5). Table 1 lists the details results of the top five processes of the GO and KEGG enrichment analysis.

## PPI network construction and identification of hub genes

The PPI network (the minimum required interaction score set to 0.15) of the overlapping genes (38 nodes, 264 edges) were obtained using Cytoscape (Fig 6A) to investigate the interaction between the proteins corresponding to the overlapping genes. The larger the gene degree value, the larger the point will be. The darker the gene color and the wider the edge, the stronger the evidence for the interaction between proteins will be. The 10 topological methods of the CytoHubba plug-in in Cytoscape were adopted to screen top10 hub genes (i.e., SOX2, KLF4, ITGB4, CD44, MSX1, YAP1, AQP1, EGR2, YWHAZ, and TFAP2C) (Fig 6B, 10 nodes, 36 edges). Table 2 lists the gene symbols, description, and functions.

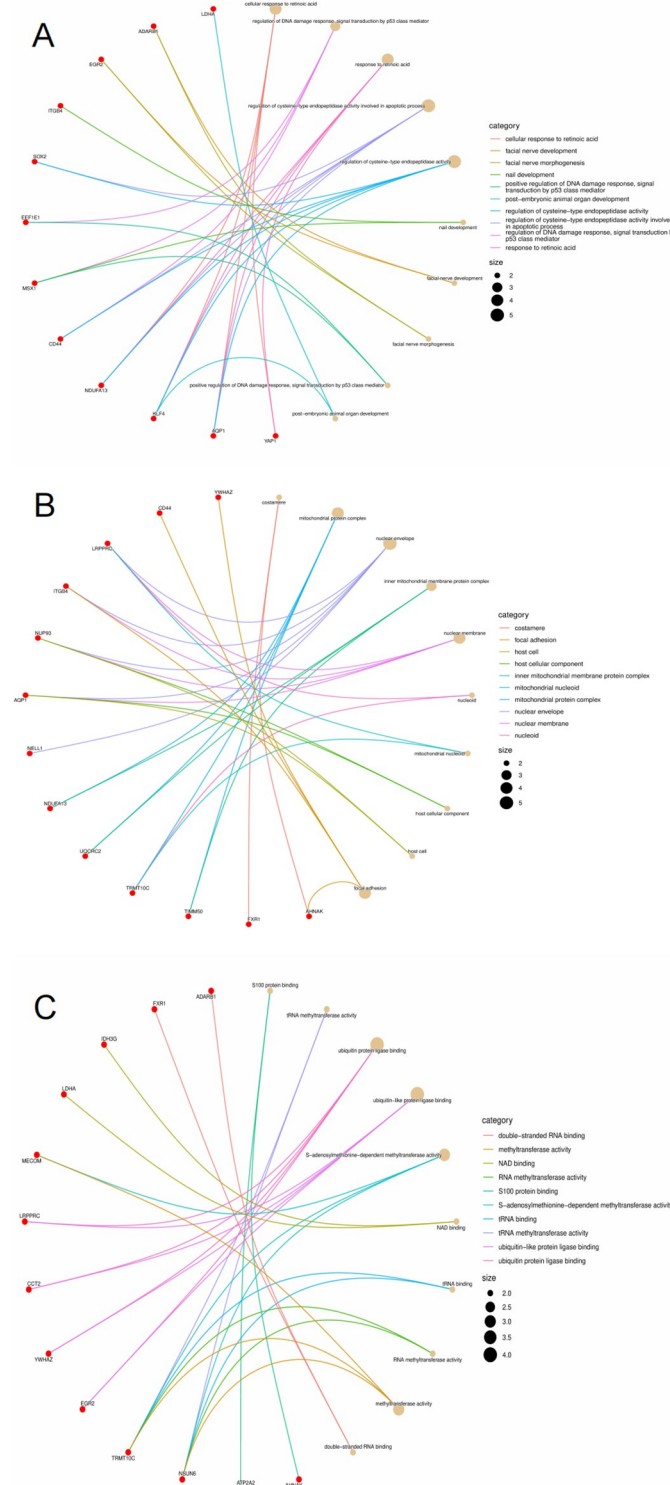

**Fig 4. Circle graph in GO enrichment of the overlapping genes of AD DGEs and m6A gene.** (A-C) The circle graph shows the overlapping genes enriched in the Top10 GO categories of BP, CC, and MF, respectively. The Go category is represented by a yellow point, the color of the line passed by the point represents the category indicated in the diagram, and the size of the point indicates the number of genes it contains.

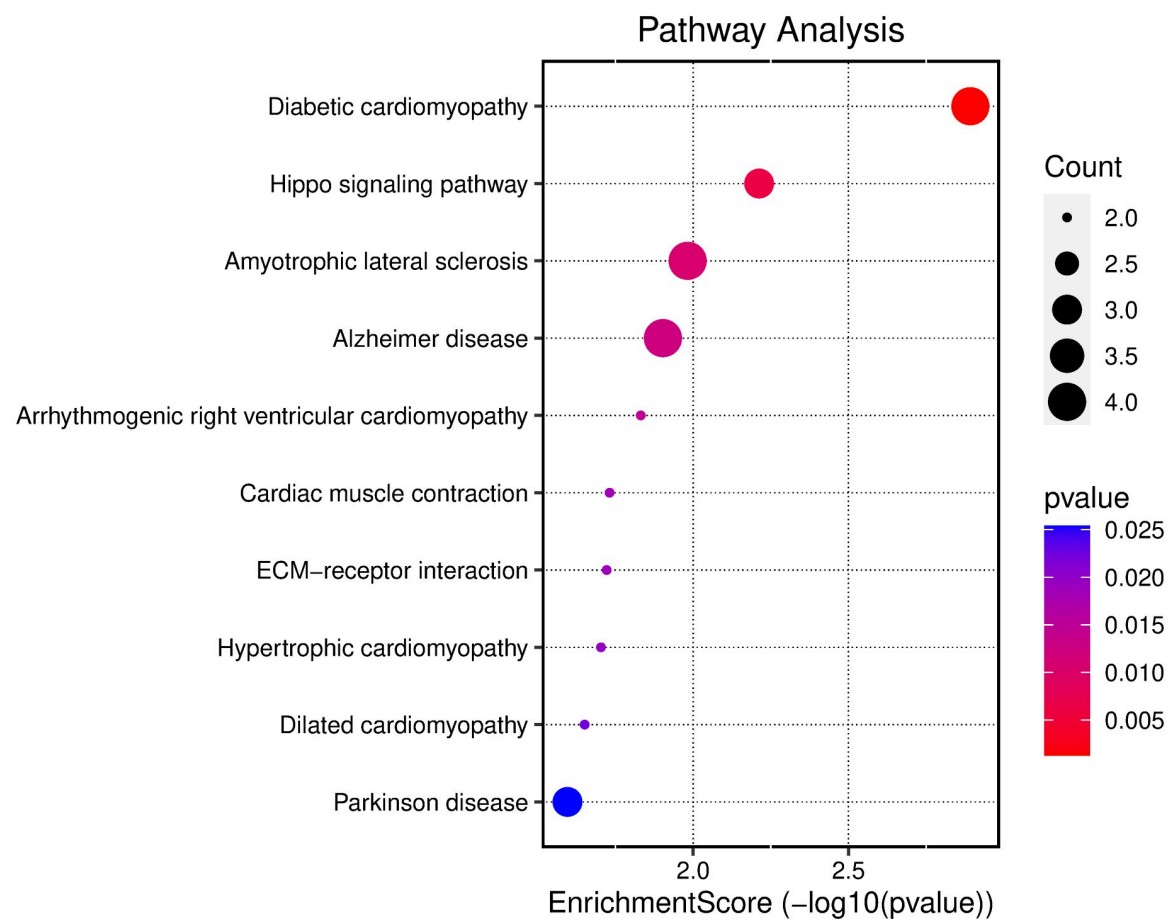

**Fig 5. KEGG enrichment analysis of the overlapping genes.** KEGG enrichment analysis of the overlapping genes of overlapping genes.

## The protein level of hub gene is expressed in different regions of human brain

The samples in GSE5281 and GSE48350 were taken from the human brain tissues. To verify the expression of central genes in different regions of the brain, we compared gene expression levels of hub genes between Cerebral cortex, Hippocampal formation, Thalamus, Hypothalamus, Midbrain, Cerebellum, and White matter in human brain tissues with the HPA database (Fig 7A). As a preliminary reference, the differential expression regions of the above-mentioned genes may exist during the development of AD. The results suggested that hug genes expressed differently in different brain regions. The expression of AQP1 in White matter was the most significant, reaching 432.4 nTPM. However, the expression level ranged from 79.1 nTPM to 181.1 nTPM in other brain regions. The expression levels of YWHAZ in Cerebral cortex and Hippocampal formation reached 343 nTPM and 366.2 nTPM, respectively. The expression levels of CD44 in Hypothalamus, Midbrain and White matter were higher than those in other subregions, ranging from 102.3 nTPM to 116.5 nTPM. The expression levels of SOX2 and KLF4 were between 46.5 nTPM and 104.8 nTPM. The expression levels of MSX1 and YAP1 in different brain regions ranged from 11.8 nTPM to 30.2 nTPM. KLF4 expression, EGR2 and TFAP2C was lower than 10 nTPM.

**Table 1. The top 5 processes with results of GO and KEGG enrichment analysis.**

| | ID | Description | pvalue | geneID | Count |
|---|---|---|---|---|---|
| BP | GO:0071300 | cellular response to retinoic acid | 9.79066E-06 | YAP1/AQP1/KLF4/NDUFA13 | 4 |
| BP | GO:0043516 | regulation of DNA damage response, signal transduction by p53 class mediator | 4.32091E-05 | CD44/MSX1/EEF1E1 | 3 |
| BP | GO:0032526 | response to retinoic acid | 6.45273E-05 | YAP1/AQP1/KLF4/NDUFA13 | 4 |
| BP | GO:0043281 | regulation of cysteine-type endopeptidase activity involved in apoptotic process | 6.77998E-05 | CD44/AQP1/KLF4/NDUFA13/SOX2 | 5 |
| BP | GO:2000116 | regulation of cysteine-type endopeptidase activity | 0.000111652 | CD44/AQP1/KLF4/NDUFA13/SOX2 | 5 |
| CC | GO:0098798 | mitochondrial protein complex | 0.001866338 | TIMM50/TRMT10C/UQCRC2/ NDUFA13 | 4 |
| CC | GO:0005635 | nuclear envelope | 0.002366697 | NELL1/AQP1/NUP93/ITGB4/LRPPRC | 5 |
| CC | GO:0098800 | inner mitochondrial membrane protein complex | 0.002608254 | TIMM50/UQCRC2/NDUFA13 | 3 |
| CC | GO:0031965 | nuclear membrane | 0.002959294 | AQP1/NUP93/ITGB4/LRPPRC | 4 |
| CC | GO:0005925 | focal adhesion | 0.009143824 | CD44/AHNAK/YWHAZ/ITGB4 | 4 |
| MF | GO:0031625 | ubiquitin protein ligase binding | 0.00321757 | EGR2/YWHAZ/CCT2/LRPPRC | 4 |
| MF | GO:0044389 | ubiquitin-like protein ligase binding | 0.004014383 | EGR2/YWHAZ/CCT2/LRPPRC | 4 |
| MF | GO:0008757 | S-adenosylmethionine-dependent methyltransferase activity | 0.004386642 | NSUN6/TRMT10C/MECOM | 3 |
| MF | GO:0008173 | RNA methyltransferase activity | 0.008975213 | NSUN6/TRMT10C | 2 |
| MF | GO:0008168 | methyltransferase activity | 0.010271748 | NSUN6/TRMT10C/MECOM | 3 |
| KEGG | hsa05415 | Diabetic cardiomyopathy | 0.001280589 | ATP2A2/UQCRC2/PDK4/NDUFA13 | 4 |
| KEGG | hsa04390 | Hippo signaling pathway | 0.006131771 | YAP1/YWHAZ/SOX2 | 3 |
| KEGG | hsa05014 | Amyotrophic lateral sclerosis | 0.010414549 | PSMA5/NUP93/UQCRC2/NDUFA13 | 4 |
| KEGG | hsa05010 | Alzheimer disease | 0.012509826 | PSMA5/ATP2A2/UQCRC2/NDUFA13 | 4 |
| KEGG | hsa05012 | Parkinson disease | 0.025391473 | PSMA5/UQCRC2/NDUFA13 | 3 |

Subsequently, the gene expression scores of hub genes in different regions of brain tissue were compared using the data originating from the BGEE database (Fig 7B). The brain regions are presented as follows: Hippocampal formation, Thalamus, Hypothalamus and Cerebellum. The results suggested the high and low gene expression scores of the above-described 10 hub genes in different brain regions, whereas the average and median gene expression scores reached 79.21 and 80.08, respectively. The above-mentioned hub genes were highly expressed in the brain regions studied. As revealed by the above result, the effect of m6A on AD may occur in the brain regions studied.

As indicated by the detailed data, the expression scores of SOX2, ITGB4, CD44, AQP1, and YWHAZ were higher in the respective brain region, and the expression scores of YWHAZ were higher than those of other genes, with an average expression score of 99.22. Moreover, SOX2 achieved higher expression scores in the Thalamus and Hypothalamus region, and ITGB and AQP1 achieved higher expression scores in the Hypothalamus region (both expression scores greater than 90). The data of expression scores of some genes in the database were missing. The existing data suggested that the expression scores of KLF4, EGR2, and TFAP2C in the brain tissue were all lower than 70. The expression scores of MSX1 and YAP1 ranged from 62.69 to 77.58, and no significant difference was identified between groups.

As indicated by the results of the expression of hug genes in different brain tissue regions in HPA and BGEE databases, YWHAZ and AQP1 were highly expressed in all brain tissue subregions. AQP1 was more significantly expressed in White matter, whereas YWHAZ was more highly expressed in Cerebral cortex and Hippocampal formation. SOX2, ITGB4, and CD44 were highly expressed in White matter and Hypothalamus regions.

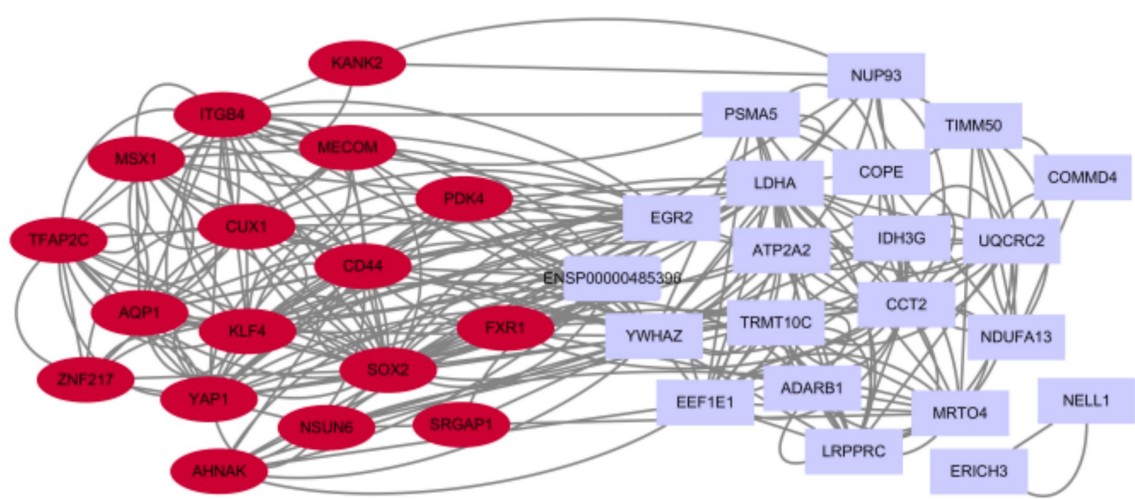

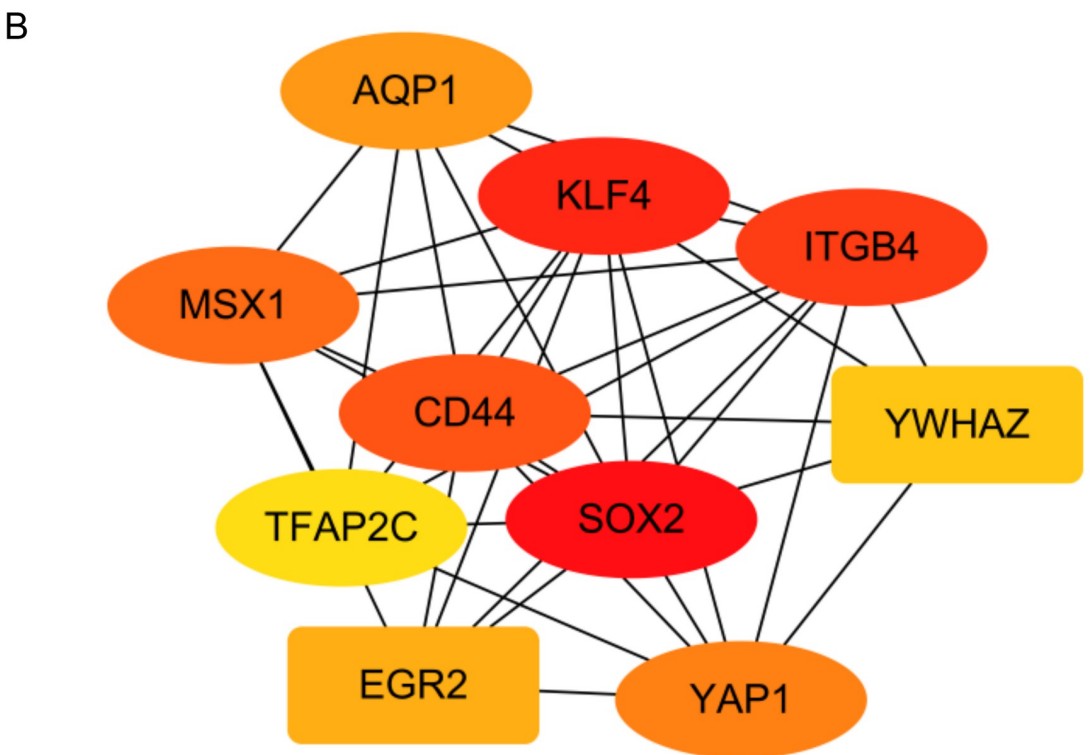

**Fig 6. Construction of PPI network of the overlapping genes and screening of hub genes.** (A) The PPI network of the overlapping genes of AD DGEs and m6A gene (38 nodes, 264 edges). The larger the gene degree value, the larger the point, and the darker the gene color. (B)The Cytohubba is used to construct the Top10 hub genes. The figure presents the Top10 hub genes built by the MCC method (10 nodes, 36edges).

**Table 2. Top 10 hub genes and their function.**

| Gene symbol | description | Function |
|---|---|---|
| SOX2 | SRY-box transcription factor 2 | Downstream SRRT target that mediates the promotion of neural stem cell self-renewal (By similarity); Keeps neural cells undifferentiated by counteracting the activity of proneural proteins and suppresses neuronal differentiation (By similarity). |
| KLF4 | Krueppel-like factor 4 | Transcription factor; can act both as activator and as repressor. Binds to the promoter region of its own gene and can activate its own transcription. |
| ITGB4 | Integrin subunit beta 4 | Integrin alpha-6/beta-4 is a receptor for laminin.ITGB4 binds to NRG1 (via EGF domain) and this binding is essential for NRG1-ERBB signaling |
| CD44 | CD44 molecule | Engages, through its ectodomain, extracellular matrix components and serves as a platform for signal transduction by assembling, via its cytoplasmic domain, protein complexes containing receptor kinases and membrane proteases. |
| MSX1 | Msh homeobox 1 | Acts as a transcriptional repressor. May play a role in limb-pattern formation. Acts in cranofacial development and specifically in odontogenesis. |
| YAP1 | Yes1 associated transcriptional regulator | Transcriptional regulator which can act both as a coactivator and a corepressor and is the critical downstream regulatory target in the Hippo signaling pathway that plays a pivotal role in organ size control. |
| AQP1 | Aquaporin-1 | Forms a water-specific channel that provides the plasma membranes of red cells and kidney proximal tubules with high permeability to water, thereby permitting water to move in the direction of an osmotic gradient. |
| EGR2 | Early growth response 2 | Sequence-specific DNA-binding transcription factor; Plays a role in hindbrain segmentation by regulating the expression of a subset of homeobox containing genes and in Schwann cell myelination by regulating the expression of genes involved in the formation and maintenance of myelin |
| YWHAZ | Tyrosine 3-monooxygenase/ tryptophan 5-monooxygenase activation protein zeta | Adapter protein implicated in the regulation of a large spectrum of both general and specialized signaling pathways. |
| TFAP2C | Transcription factor AP-2 gamma | Sequence-specific DNA-binding protein that interacts with inducible viral and cellular enhancer elements to regulate transcription of selected genes. |

## Identification and verification of hub genes

Box plots and Scatter plot of GSE5281 and GSE48350 gene expression levels showed that hub genes expression levels were significantly different between AD and control groups (Fig 8A–8D). In the GSE5281 dataset, except KLF4, ITGB4 and CD44, there were significant differences in genes expression between the control and AD groups ($p < 0.01$). In GSE48350 dataset, except for ITGB4 and MSX1, there were significant differences in genes expression between control and AD groups ($p < 0.05$).

The diagnostic value of the 10 core genes in AD patients was assessed by using receiver operating characteristic curves, which were mutually validated in the GSE5281 and GSE48350 datasets, respectively. The AUC of all hub genes were greater than 0.500. In the GSE5281 dataset (Fig 9A), YAP1 (AUC, 0.839) had the highest value, followed by MSX1 (AUC, 0.794), TFAP2C (AUC, 0.732) and SOX2 (AUC, 0.712). In the GSE48350 dataset (Fig 9B), YWHAZ (AUC, 0.706) had the highest value, followed by SOX2 (AUC, 0.664), CD44 (AUC, 0.662), YAP1 (AUC, 0.638), and EGR2 (AUC, 0.636).

## Screening for potential pharmacological targets

Prediction of potential pharmacological targets of hub genes in the cMAP database. The compounds were sorted and then screened in accordance with their scores. The first 10 drugs suggested for AD are LDN-193189, sarmentogenin, cucurbitacin-i, KI-8751, SCH-79797, AG-592, YM-155, BAX-channel-blocker, SN-38 and cercosporin (Table 3). The above-described compounds may play a therapeutic role in the course of AD.

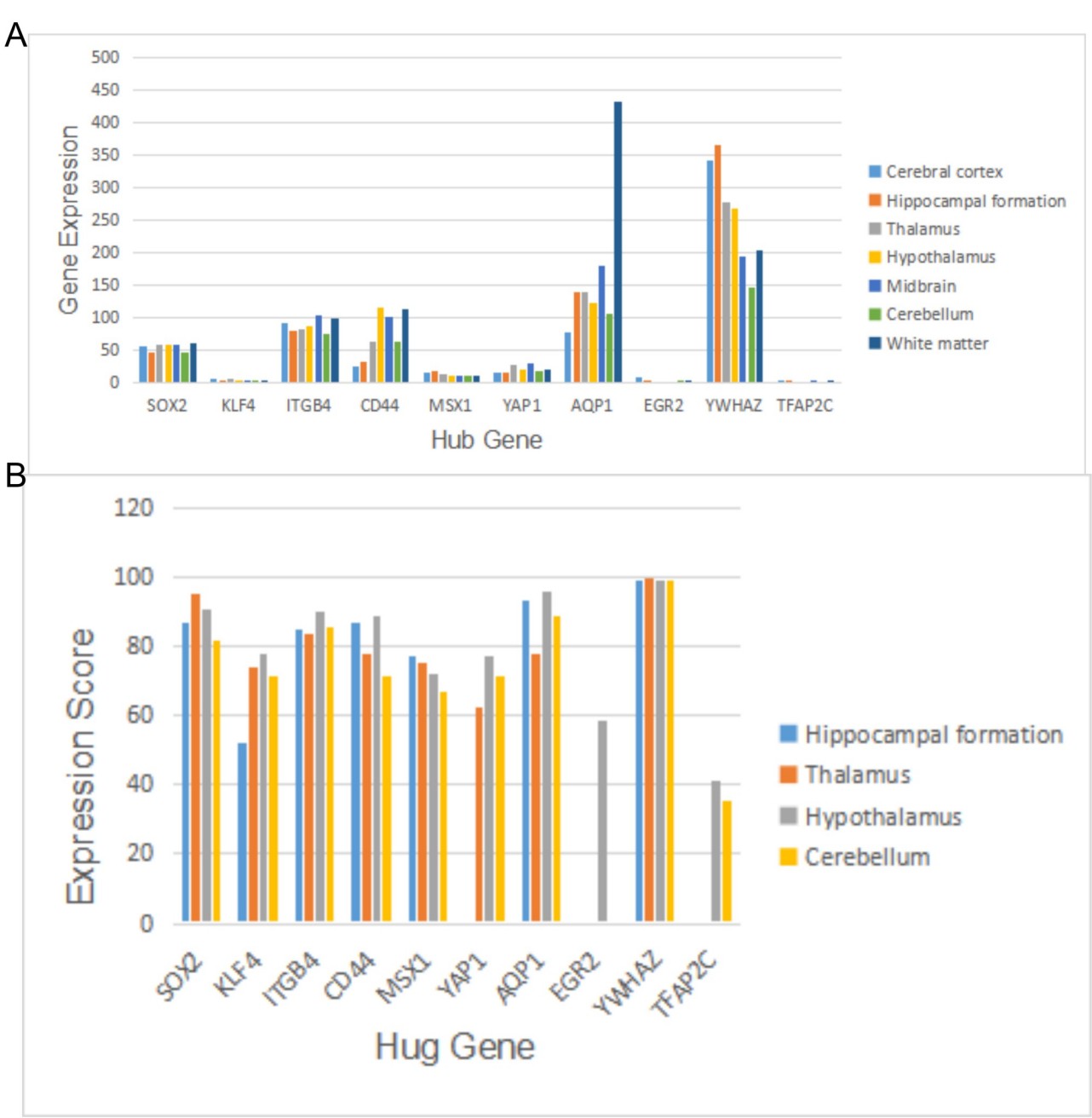

**Fig 7. The expression analysis of hub genes in human brain tissues.** (A) The analysis and summary of gene expression of hub genes in human brain tissues from the HPA database. (B) The gene expression scores of hub genes in different regions of the human brain tissues were compared using the Bgee database. A high score represents the gene expressed at high levels in this regions.

## Discussion

Alzheimer's disease refers to a neurodegenerative disorder that generally results in memory impairment and cognitive impairment [33]. Most existing research has reported that the pathological changes of AD arise from the accumulation of amyloid β plaques in the brain, mainly in the form of inflammation and tau protein aggregation in neurofibrillary tangles [34]. Several

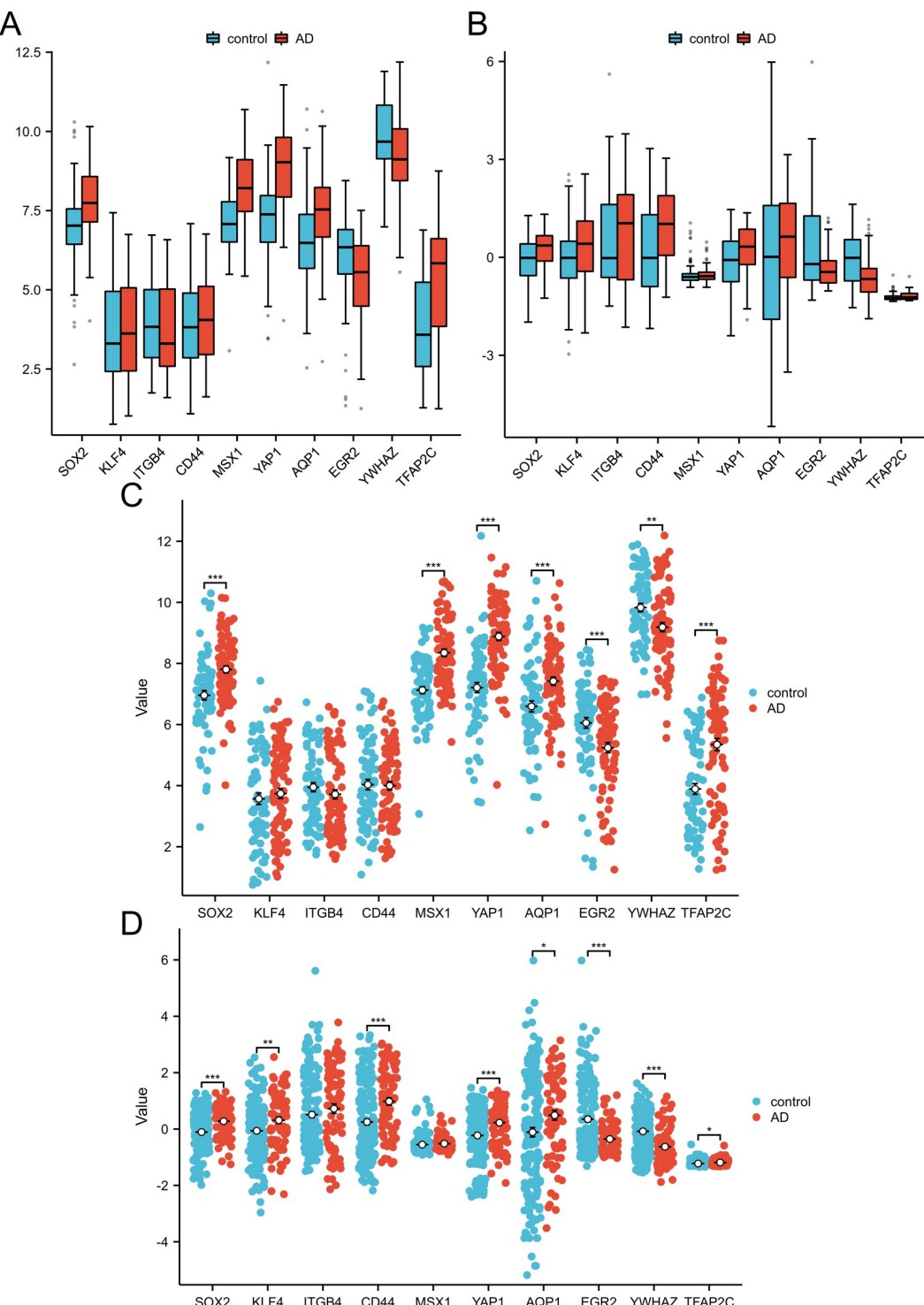

**Fig 8. Comparison of the expression of 10 hub genes in AD group and normal control group.** (A) Comparative expression of hub genes in the GSE5281 dataset. (B) Comparative expression of hub genes in the GSE48350 dataset. (C) Scatter plots of 10hub genes expression in AD and control groups were compared in GSE5281 dataset. (D) Scatter plots of 10hub genes expression in AD and control groups were compared in GSE48350 dataset. *p <0.05; **p < 0.01; ***p < 0.001.

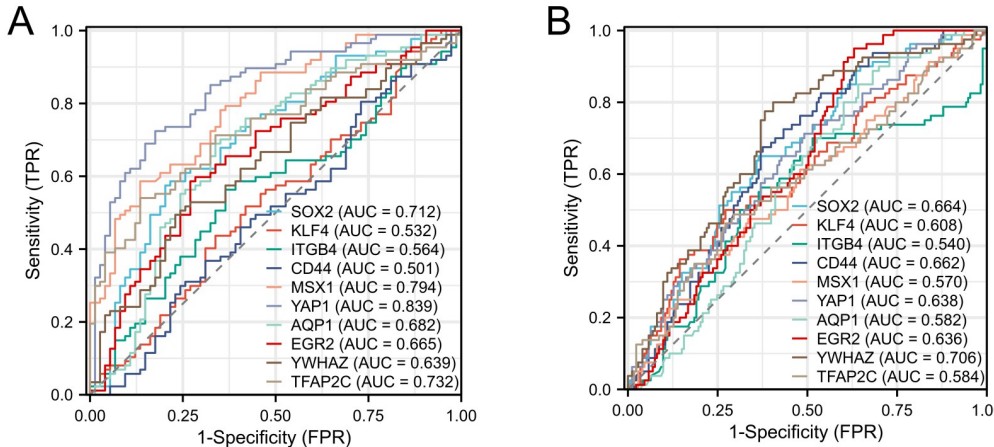

**Fig 9. ROC curves of 10 hub genes.** (A) Diagnostic value of hub genes in the GSE5281 dataset. (B) Diagnostic value of hub genes in the GSE48350 dataset.

studies have demonstrated that m6A gene mutation and high level of gene expression show a correlation with the occurrence and development of AD [21, 22]. However, the mechanism of action remains unclear and should be studied in depth.

In this study, the brain tissue samples were first collected from AD patients from public databases, and the differentially expressed genes of m6A in AD patients' brain tissues were screened based on the analysis result of the gene expression profiling of the samples. A total of 42 DEGs were found, including 20 up-regulated genes and 22 down-regulated genes. This proves that m6A plays a role in the development of AD.

Subsequently, we performed GO and KEGG enrichment analyses on DGEs. The results of the GO analysis showed that the overlapping genes were mainly enriched in involving in retinoic acid, DNA damage response and cysteine-type endopeptidase activity of biological processes. This is consistent with previous studies. There is evidence that many steps in the retinoic acid signaling pathway, including the decline of binding proteins and metabolic enzymes, are involved in the development of AD. And there are some studies show that a

**Table 3. Top 10 prediction results from cMap for AD.**

| Score | Name | Description |
|---|---|---|
| -99.71 | LDN-193189 | ALK inhibitor, serine/threonine protein kinase inhibitor |
| -98.47 | sarmentogenin | ATPase inhibitor |
| -98.37 | cucurbitacin-i | inhibitor of STAT3/JAK2 signaling, JAK inhbitor, lipocortin synthesis stimulant, STAT inhibitor |
| -98.15 | KI-8751 | PDGFR alpha and c-Kit inhibitor, vascular endothelial growth factor receptor 2 (VEGFR2) inhibitor, VEGFR inhibitor |
| -98.03 | SCH-79797 | proteinase activated receptor antagonist |
| -97.53 | AG-592 | tyrosine kinase inhibitor |
| -97.52 | YM-155 | survivin inhibitor, XIAP expression inhibitor |
| -97.37 | BAX-channel-blocker | cytochrome C release inhibitor |
| -97.31 | SN-38 | topoisomerase inhibitor |
| -97.26 | cercosporin | photoactivated to produce toxic reactive oxygen species, PKC inhibitor |

sustained decline in the mRNA and protein of retinoic acid receptors is aberrant with a decline in retinoic acid receptor β and γ transcripts at the early stages of the AD model [35, 36]. Existing research has suggested that persistent DNA damage response is a major driver of the aging phenotype, thus accelerating neuronal cell senescence and causing AD [37]. As indicated by the results of the GO analysis, the mitochondrial protein complexes were mainly enriched in involving in cellular component. Some Previous research has suggested that mitochondrial complexes are capable of inhibiting the development of AD by inducing beneficial mitochondrial stress responses [38]. A growing body of research has confirmed that RNA methylation affects neural development and aging, and dysmethylation of RNA can directly cause the neurodegenerative lesions that comprise AD [21, 39], consistent with the Molecular Function results of DGEs enrichment in this study. The result of the KEGG pathway analysis suggested that the DEGs in AD are primarily involved in Hippo signaling pathway. The Hippo signaling pathway is a mechanosensory pathway in microglia, acting through mechanization and subsequent protein kinase cascades that take on critical significance to affecting neuronal development and many other cellular processes. Hippo signaling pathway may serve as a potential therapeutic target for the prevention of microglia-induced neurodegeneration in AD [40, 41].

In addition, a protein-interaction network of m6A and AD intersecting genes was built by STRING, the PPI network was visualized using Cytoscape software, and 10 hub genes (i.e., SOX2, KLF4, ITGB4, CD44, MSX1, YAP1, AQP1, EGR2, YWHAZ, and TFAP2C) were screened using CytoHubba plug-in. SOX2 deficiency can result in neurodegeneration and injured neurons in the model rat brain. Moreover, previous research has indicated that SOX2 neural stem cells exhibit multifunction and self-renewal ability in the hippocampus of the adult model [42, 43]. KLF4 plays a vital regulatory role in the neurophysiological and neuropathological processes of AD [44]. KLF4 is highly expressed in the mouse model of AD. Existing research has suggested that KLF4 expression is positively correlated with extracellular deposition of amyloid-β peptide, such that AD is caused [45]. Accordingly, KLF4 may serve as a potential therapeutic target for AD. CD44, a marker for a subset of astrocyte, can be increased in the frontal cortex of AD patients, as indicated by previous research. The astrocyte also responds to neurofibrillary tangle and plaques, tau and AB in Hyperphosphorylation, and it may exert neuroprotective or deleterious effects [46]. In a weighted gene co-expression network analysis, MSX1 is highly correlated with AD phenotype in a study of key genes in Alzheimer disease, and the aged AD transgenic mice have the up-regulated expression of MSX1 [47]. Yap1 has been confirmed as one of the pivotal genes in Alzheimer's disease. It plays a certain role in upstream regulation and may prevent astrocyte aging in AD models via the CDK6 signaling pathway, such that cognitive decline can be slowed down [48, 49]. The high expression of AQP1 may impair the cognitive function of the AD model by inhibiting the Wnt signaling pathway and facilitating neuronal apoptosis. In contrast, AQP1 silencing is capable of protecting the hippocampal neurons of the AD mouse model and improving the cognitive function of AD mice [50]. As indicated by the results of existing research, EGR2 expression in the hippocampus is significantly increased in the aging process, which may play a certain role in the biological process of immune response to external stimuli [51]. Previous research has suggested that YWHAZ is directly correlated with AD. YWHAE is significantly altered in the mouse model of AD, and it may serve as a key regulatory gene in the prefrontal cortex of AD [52]. In general, the above-mentioned 10 hub genes are potential targets for the treatment of AD.

In brief, in this study, the differentially expressed genes in AD were screened, and the overlapping genes associated with DGEs and m6A genes were identified. A total of 42 DEGs and 10 hub genes were screened. The biological changes may be correlated with retinoic acid, DNA damage response, and cysteine-type endopeptidase activity, which may occur via the Hippo signaling pathway. In this study, human brain tissue data were employed, more

consistent with the brain tissue gene expression of AD patients. Based on the research on the gene expression of 10 hub gene in different brain regions, a direction can be provided for investigating the location of m6A gene aberrant mutation in the development of AD, so as to predict potential therapeutic compounds based on hub gene and lay a theoretical basis for the in-depth study of AD therapy. Some limitations remained in this study. The differential expression, mechanism and pathway of hub gene in Alzheimer's disease, and the potential therapeutic compounds should be verified through further biological model experiments and using clinical samples.

## Conclusion

In general, the potential role of m6A modification in Alzheimer's disease was investigated. The relevant hub genes (e.g., SOX2, KLF4, ITGB4, CD44, MSX1, YAP1, AQP1, EGR2, YWHAZ, and TFAP2C) and the involved pathways were identified through the bioinformatics analysis. The biological changes may be correlated with retinoic acid, DNA damage response, and cysteine-type endopeptidase activity, which may occur via the Hippo signaling pathway. Furthermore, the brain regions with high expression of hub gene were identified, and the compounds exerting potential therapeutic effects on AD were predicted. On that basis, this study can lay a theoretical basis for exploring novel therapies for AD.

## Acknowledgments

The authors gratefully acknowledge the data provided by patients and researchers participating in GEO.

## Author Contributions

**Data curation:** Pengyun Ni, Kaiting Pan.

**Formal analysis:** Pengyun Ni, Kaiting Pan, Bingbing Zhao.

**Methodology:** Pengyun Ni.

**Software:** Bingbing Zhao.

**Writing – original draft:** Pengyun Ni.

**Writing – review & editing:** Pengyun Ni.

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
