## [Decision Letter · Decision Letter 0]

3 Feb 2023

PONE-D-22-31888Influence of N6-Methyladenosine (m6A) Modification on Cell Phenotype in Alzheimer’s DiseasePLOS ONE

Dear Dr. Ni,

Thank you for submitting your manuscript to PLOS ONE. After careful consideration, we feel that it has merit but does not fully meet PLOS ONE’s publication criteria as it currently stands. Therefore, we invite you to submit a revised version of the manuscript that addresses the points raised during the review process.

We look forward to receiving your revised manuscript.

Kind regards,

Divakar Sharma

Academic Editor

PLOS ONE

Journal Requirements:

- https://www.frontiersin.org/articles/10.3389/fimmu.2021.753929/full?

In your revision ensure you cite all your sources (including your own works), and quote or rephrase any duplicated text outside the methods section. Further consideration is dependent on these concerns being addressed. 

  "NO. The funders had no role in study design, data collection and analysis, decision to publish, or preparation of the manuscript."

Reviewers' comments:

Reviewer's Responses to Questions

**Comments to the Author**

1. Is the manuscript technically sound, and do the data support the conclusions?

Reviewer #1: Partly

Reviewer #2: Partly

2. Has the statistical analysis been performed appropriately and rigorously? 

Reviewer #1: No

Reviewer #2: I Don't Know

3. Have the authors made all data underlying the findings in their manuscript fully available?

Reviewer #1: Yes

Reviewer #2: Yes

4. Is the manuscript presented in an intelligible fashion and written in standard English?

Reviewer #1: Yes

Reviewer #2: Yes

5. Review Comments to the Author

Reviewer #1: In this study, the authors studied the N6-Methyladenosine modification related differentially expressing genes in Alzheimer’s disease using available dataset from public database. The authors identified biological processes and molecular functional pathways that are affected in the patient samples group. Further, the authors tested the pharmacological targets using cMAP database. The authors derived their conclusions using only one dataset and since this is basically a bioinformatics study, the authors should test multiple datasets for better conclusions. In addition, the authors must further clearly describe their crucial findings on biological functions that benefits the field in abstract and other relevant sections clearly.

Major Comments

• Introduction section should be expanded in few sentences stating about how mechanistically m6A modification occurs in mammals. Also, a sentence about demethylation.

• In fig. 1 and 2, the authors showed DEGs and m6A associated genes. However, the authors should also show the differences in genes associated with hyper and hypo methylation.

• Have the authors attempted to study DEGs and GO (figs 1-5) in the brain tissue specific manner? It would be interesting to study tissue specific differences and the influences of m6A modifications.

• Why the authors used only one dataset, GSE5281 in their study. The authors should include multiple available datasets which have different sources of tissue samples for a better interpretation.

• Have the authors attempted to check statistical significance in fig 7?

Minor Comments

• Indicate the threshold cut-off used for PPI analysis.

• The authors should make a table for top 10 genes from KEGG. The authors should revisit fig. 3 legend as I don’t see dot plot in fig. 3.

Reviewer #2: Dear Editor,

The manuscript titled “Influence of N6-Methyladenosine (m6A) Modification on Cell Phenotype in Alzheimer’s Disease’’ tries to understand the role of methyl adenosine in Alzheimer’s disease using insilico biology. “This manuscript needs the following major revision before being accepted”. Also please be informed that, I have limited knowledge in bioinformatics databases and the methodologies used needs a validation from bioinformatician.

Comments

1. Overall English could be improved

2. we found 1277 differently expressed genes in brain tissues of AD patients compared with normal brain tissues. Of these, 498 were up-regulated and 779 were down-regulated. A recent article using the GSE5281 2,063 DEGs comprising 854 upregulated genes and 1,209 downregulated genes were obtained based on microarray analysis of GSE5281 ( https://www.karger.com/Article/Abstract/518727) . Why there exists discrepancy in DEG’s

3. Can the cross -genes could be renamed as overlapping genes as the name seems to be confusing.

4. Using the venn diagram method the authors integrated differential expression data from GSE5281 and GeneCards database. Whether the presence of m6A modifications were validated in the overlapping genes which formed the basis for their model.

5. It would be interesting to show m6A modifications using sequencing data atleast for two overlapping genes atleast supporting data

6. PLOS authors have the option to publish the peer review history of their article (what does this mean?). If published, this will include your full peer review and any attached files.

Reviewer #1: No

Reviewer #2: No

---

## [Author Response · Author response to Decision Letter 0]

19 Apr 2023

Dear reviewers,

We thank you very much for giving us an opportunity to revise our manuscript. We appreciate editor and reviewers very much for your positive and constructive comments and suggestions on our manuscript. The comments and suggestions are all valuable and very helpful for revising and improving our paper, as well as the important guiding significance to our researches. We have studied comments carefully and have made correction which we hope meet with approval.

In this revised version, we have addressed the concerns of the reviewers. An point-by-point response to the reviewers' comments is enclosed. We hope that these revisions successfully address their concerns and requirements and that this manuscript will be accepted. 

Best wishes,

Pengyun Ni

Reviewer #1: 

In addition, the authors must further clearly describe their crucial findings on biological functions that benefits the field in abstract and other relevant sections clearly.

Response：As you suggested, we have described biological functions and signaling pathways in the abstract, conclusions, and discussion sections.

Major Comments

• Introduction section should be expanded in few sentences stating about how mechanistically m6A modification occurs in mammals. Also, a sentence about demethylation.

Response：As you suggested, we have added in the introduction a key mechanism by which M6A modification occurs in mammals.

•In fig. 1 and 2, the authors showed DEGs and m6A associated genes. However, the authors should also show the differences in genes associated with hyper and hypo methylation.

Response：As you suggested, we have identified some of the cross-expressed overlopping genes for DEGs and m6A in Fig 1. Red colour indicates up-regulated genes, and blue indicates down-regulated genes. In addition, we show the expression of up-regulated and down-regulated genes between AD and control groups by PCA. PCA shows that the Control group and AD group samples were separated, indicating that there were differences in gene expression of up-regulated and down-regulated overlapping genes different between the groups.

• Have the authors attempted to study DEGs and GO (figs 1-5) in the brain tissue specific manner? It would be interesting to study tissue specific differences and the influences of m6A modifications.

Response：I’m so sorry that our team does not have the equipment and reagents to do translational research. Because our hospital belong to the prefecture-level City Hospital of traditional Chinese medicine which is based on clinical work and clinical research. In addition, this research has not been supported by any funds. It does not have the enough fund for further experimental validation study. If there are sufficient conditions to support our further supplement experimental study, we would validate the mechanisms of action and the signaling pathways of molecules and protein targets in the findings of this study.

• Why the authors used only one dataset, GSE5281 in their study. The authors should include multiple available datasets which have different sources of tissue samples for a better interpretation.

Response：Our initial idea for this study was to study human specimens, as this is more in line with the actual state of the patient. Therefore, in the selection of the data set, the species is restricted to homo. Through your suggestion, We realized the imprecision of using only one data set. We then consulted an experienced teacher who suggested that it would be more scientific to select a data set from the same platform. In addition, in order to ensure sufficient sample size and valid data, we select data sets with sample size greater than 50.

So our criteria for selecting the dataset were: the disease name was“Alzheimer's disease”, the species was “homo sapiens”, GPL570 (the same platform as GSE5281) , and valid data were available for GEO2R analysis. Finally, we selected GSE5281 and GSE48350 as the data sources in the GPL570 platform.

• Have the authors attempted to check statistical significance in fig 7?

Response：The data in Fig 7 are derived from the HPA (Human Protein Atlas) database and the BGEE database. The data in these two databases were obtained by counting the corresponding human tissue specimens. In the database, we can only see the average value of gene expression in tissues, and the standard deviation or error value can not be collected, so we did not do statistical significance analysis. Only by describing and analyzing the average of gene expression, we can verify the expression of the core genes in each brain region.

Minor Comments

• Indicate the threshold cut-off used for PPI analysis.

Response：As you suggested, we added a threshold cut-of for PPI.

• The authors should make a table for top 10 genes from KEGG. The authors should revisit fig. 3 legend as I don’t see dot plot in fig. 3.

Response：As you suggested, we have supplemented the tables for the results of the GO and KEGG analyses and modified the legend for Fig. 3.

Reviewer #2:

1.Overall English could be improved.

Response：We asked experienced teacher to polish our paper and carefully checked the spelling, grammar and formatting. And we also asked professional editors to review our articles for revisions. We believe the revised version will meet the English presentation standard.

2.We found 1277 differently expressed genes in brain tissues of AD patients compared with normal brain tissues. Of these, 498 were up-regulated and 779 were down-regulated. A recent article using the GSE5281 2,063 DEGs comprising 854 upregulated genes and 1,209 downregulated genes were obtained based on microarray analysis of GSE5281 ( https://www.karger.com/Article/Abstract/518727) . Why there exists discrepancy in DEG’s.

Response：We noticed in the materials and methods section of that article showed that"We identified DEGs by using the Limma package in R software (version 3.5.2) . Log2FC >1 and FDR ≤0.05”. At the time, we used the GSE5281 data set to filter DEGs based on“|log2(Fold Change)| > 1 and P.value < 0.05 “ and used the Limma package in R software (version 4.2.1) .

On the one hand, two articles had different screening conditions for DEGs. On the other hand, the two articles use different versions of R software. In some GEO data analysis lectures, the teachers mentioned that different computers and R software screen DEGs because the background parameters of the computers and R software are not set in the same way, even with the same data set and the same filter conditions, the resulting DEGs will be different and can not be exactly the same.

3.Can the cross -genes could be renamed as overlapping genes as the name seems to be confusing.

Response：We have modified it according to your suggestion.

4.Using the venn diagram method the authors integrated differential expression data from GSE5281 and GeneCards database. Whether the presence of m6A modifications were validated in the overlapping genes which formed the basis for their model.

Response：I’m so sorry that our team does not have the equipment and reagents to do the verification experiment.. Because our hospital belong to the prefecture-level City Hospital of traditional Chinese medicine which is based on clinical work and clinical research. In addition, this research has not been supported by any funds. It does not have the enough fund for further experimental validation study. If there are sufficient conditions to support our further supplement experimental study, we would validate the mechanisms of action and the signaling pathways of molecules and protein targets in the findings of this study.

5.It would be interesting to show m6A modifications using sequencing data atleast for two overlapping genes atleast supporting data.

Response：As you suggest, we consider that it is more scientific to choose a data set on the same platform. In addition, in order to ensure sufficient sample size and valid data, we select data sets with sample size greater than 50. So our criteria for selecting the dataset were: the disease name was“Alzheimer's disease”, the species was “homo sapiens”, GPL570 (the same platform as GSE5281) , and valid data were available for GEO2R analysis. Finally, we selected GSE5281 and GSE48350 as the data sources in the GPL570 platform.

---

## [Decision Letter · Decision Letter 1]

27 Jun 2023

PONE-D-22-31888R1Influence of N6-Methyladenosine (m6A) Modification on Cell Phenotype in Alzheimer’s DiseasePLOS ONE

Dear Dr. Ni,

Thank you for submitting your manuscript to PLOS ONE. After careful consideration, we feel that it has merit but does not fully meet PLOS ONE’s publication criteria as it currently stands. Therefore, we invite you to submit a revised version of the manuscript that addresses the points raised during the review process.

ACADEMIC EDITOR: Revision Required

We look forward to receiving your revised manuscript.

Kind regards,

Divakar Sharma

Academic Editor

PLOS ONE

Journal Requirements:

Additional Editor Comments:

Required the parallel scientific experimental validations of the bioinformatics data and supportive justification.

Similar concern raised by reviewer 2, comment 5.

Reviewers' comments:

Reviewer's Responses to Questions

**Comments to the Author**

1. If the authors have adequately addressed your comments raised in a previous round of review and you feel that this manuscript is now acceptable for publication, you may indicate that here to bypass the “Comments to the Author” section, enter your conflict of interest statement in the “Confidential to Editor” section, and submit your "Accept" recommendation.

Reviewer #1: All comments have been addressed

Reviewer #2: All comments have been addressed

2. Is the manuscript technically sound, and do the data support the conclusions?

Reviewer #1: Yes

Reviewer #2: Yes

3. Has the statistical analysis been performed appropriately and rigorously? 

Reviewer #1: Yes

Reviewer #2: I Don't Know

4. Have the authors made all data underlying the findings in their manuscript fully available?

Reviewer #1: Yes

Reviewer #2: Yes

5. Is the manuscript presented in an intelligible fashion and written in standard English?

Reviewer #1: Yes

Reviewer #2: Yes

6. Review Comments to the Author

Reviewer #1: (No Response)

Reviewer #2: Dear Editor,

The manuscript entitled “Influence of N6-Methyladenosine (m6A) Modification on Cell Phenotype in Alzheimer’s Disease” has incorporated possible corrections by the authors. I appreciate the authors for their efforts on this study, the analysis and data interpretations are complete bio-informatics based and does not show any experimental validations using any DNA based techniques. Similar articles published elsewhere show parallel scientific experimental validations.

My decision is to reject the article, I hope the authors would positively move ahead.

7. PLOS authors have the option to publish the peer review history of their article (what does this mean?). If published, this will include your full peer review and any attached files.

Reviewer #1: No

Reviewer #2: No

---

## [Author Response · Author response to Decision Letter 1]

1 Jul 2023

Dear Editor and reviewers,

Thank you for considering the revised version of our manuscript. We are very grateful to the editors and reviewers for your recognition and expectation of our manuscript and scientific works. We understand that the experimental verification of the conclusions of this study will be more convincing and scientific. However, the equipment, funding, and research resources currently possessed by our team cannot support our requirements for experimental verification. In our plan, if this manuscript is published, we may be able to apply for some research funding or cooperation opportunities. Thus, a series of animal experiments and cell experiments can be carried out. For example, we may be able to conduct research on the mechanism of Alzheimer's disease by upregulating or knocking out core genes in the Hippo signaling pathway, and verify the biological processes, cellular components, and molecular functions conclusions in this manuscript. I am sorry that we are currently unable to conduct relevant experimental research in our current situation. I hope to receive your understanding. 

We have checked the references cited and checked for retracted references. We have refined the format of the references. In addition, because reference no. 27 was not available on the pubmed website, we substituted references to other relevant literature. All changes made to the references section are marked in red in the revision.

Thank you again for your guidance on our research. Looking forward to hearing from you soon.

Sincerely yours,

Pengyun Ni

---

## [Editor Report · Decision Letter 2]

11 Jul 2023

Influence of N6-Methyladenosine (m6A) Modification on Cell Phenotype in Alzheimer’s Disease

PONE-D-22-31888R2

Dear Dr. Ni

We’re pleased to inform you that your manuscript has been judged scientifically suitable for publication and will be formally accepted for publication once it meets all outstanding technical requirements.

Kind regards,

Divakar Sharma

Academic Editor

PLOS ONE

Additional Editor Comments (optional):

Accept
---

## [Editor Report · Acceptance letter]

27 Jul 2023

PONE-D-22-31888R2 

Influence of N6-Methyladenosine (m6A) Modification on Cell Phenotype in Alzheimer’s Disease 

Dear Dr. Ni:

I'm pleased to inform you that your manuscript has been deemed suitable for publication in PLOS ONE. Congratulations! Your manuscript is now with our production department. 

Kind regards, 

on behalf of

Dr. Divakar Sharma 

Academic Editor

PLOS ONE